# Strain Rate Dependence and Recrystallization Modeling for TC18 Alloy during Post-Deformation Annealing

**DOI:** 10.3390/ma16031140

**Published:** 2023-01-29

**Authors:** Zhaosen Li, Jinyang Ge, Bin Kong, Deng Luo, Zhen Wang, Xiaoyong Zhang

**Affiliations:** 1State Key Laboratory of Powder Metallurgy, Central South University, Lu Mountain South Road, Changsha 410083, China; 2Hunan Xiangtou Goldsky Titanium Metal Co. Ltd., No. 116, Linyu Road, Yuelu Zone, Changsha 410221, China; 3Xiangtan Iron & Steel Group Co. Ltd., Yue Tang District, Xiangtan 411104, China

**Keywords:** TC18 alloy, strain rate, post-deformation annealing, post-dynamic recrystallization, kinetic model

## Abstract

In this paper, the dependence of dynamic recrystallization (DRX) and post-dynamic recrystallization (PDRX) of TC18 alloy on strain rate within the range of 0.001 s^−1^~1 s^−1^ was investigated through isothermal compression and subsequent annealing in the single-phase region. Electron backscatter diffraction (EBSD) characterization was employed to quantify microstructure evolution and to reveal the recrystallization mechanism. At the thermo-deformation stage, the DRX fraction does not exceed 10% at different strain rates, due to the high stacking fault energy of the β phase. During the subsequent annealing process, the total recrystallization fraction increases from 10.5% to 79.6% with the strain rate increasing from 0.001 s^−1^ to 1 s^−1^. The variations in the geometrically necessary dislocation (GND) density before and after annealing exhibit a significant discrepancy with the increasing strain rate, indicating that the GND density is a key factor affecting the PDRX rate. The PDRX mechanisms, namely meta-dynamic recrystallization (MDRX), continuous static recrystallization (CSRX) and discontinuous static recrystallization (DSRX), were also revealed during the annealing process. A new kinetic model coupling DRX and PDRX was proposed to further describe the correlation between recrystallization and the strain rate during continuous deformation and annealing. This new model facilitates the prediction of recrystallization fraction during isothermal deformation and annealing of titanium alloys.

## 1. Introduction

Near β titanium alloys are attractive materials for aerospace applications due to their high strength and good workability [1,2,3,4]. Subtransus and supertransus thermomechanical processing (TMP) are widely used for titanium alloys in order to obtain the desired microstructure and properties that meet the service conditions. The TMP process above the transus temperature can achieve a β grain refinement by breaking the as-cast microstructure. Numerous studies have shown that dynamic recovery (DRV) and dynamic recrystallization (DRX) are the main mechanisms during supertransus thermal deformation [5,6,7,8]. However, for large billets, the core temperature drops very slowly at the end of deformation, which is equivalent to experience a period of annealing. Further recrystallization occurs, for instance, meta-dynamic recrystallization (MDRX) originated from DRX grains and static recrystallization (SRX) merged by recrystallization nucleation and sub-grains rotation [9,10]. They are also collectively referred to as post-dynamic recrystallization (PDRX), that have a great influence on microstructural uniformity [11,12].

The PDRX behavior during the annealing process is influenced by the historical deformation and annealing parameters. It has been widely reported in some materials, such as aluminum alloy, Inconel, steels, etc. Zhang et al. [13] studied the PDRX behavior of 7055 aluminum alloy using a two-pass isothermal compression test, and found that the long pass interval time was beneficial for finer grain during the pass interval period. Chen et al. [14] investigated the recrystallization of 30Cr2Ni4MoV steel during hot deformation and revealed that a high temperature and larger strain rate had a strong effect on the PDRX of the material. A. Després et al. [15] found that the PDRX grains were growing rapidly because of a high strain rate during the annealing process, which could be as short as a few seconds to achieve full recrystallization in Inconel 718. Nicolaÿ et al. [16] reported that at high strain rates, the microstructure evolution after deformation was mainly dependent on MDRX. At the low strain rate of 0.001 s^−1^, SRX was shown to be the dominant mechanism.

In addition, some kinetic models were further established to describe the correlation between recrystallization and deformation conditions. Shi et al. [17] calculated the kinetic equations of PDRX by two-stage hot compression, and found that the increase of the strain rate and deformation could promote the occurrence of PDRX in LZ50 steel, which had a high fit with the simulation results of the CA method. Tang et al. [18] developed a model to characterize the microstructure evolution of TiAl alloy during PDRX using double-hit and triple-hit hot compression tests at 1150 ℃ under the strain rate of 0.001 s^−1^, and found that the thermal deformation conditions had different effects on each evolutionary mechanism during the post-deformation annealing process. Ding et al. [19,20] found that the fraction of PDRX identified by the two-stage hot compression tests was not applicable in high-level stacking fault energy materials. It could also affect the accuracy of the kinetic model in materials, such as titanium alloy. Hence, a new method is proposed based on electron backscatter diffraction with the grain orientation spread approach as a direct measurement technique to calculate the PDRX fraction and establish a kinetic model.

Up until now, the conventional understanding of PDRX behavior has mostly focused on materials with a low to medium stacking fault energy (SFE), because DRX is the main mechanism in their thermal deformation [19]. However, titanium alloys, as a medium to high SFE material, are dominated by DRV during thermal deformation, but the presence of DRX also plays a non-negligible role in microstructure evolution [7,20]. Meanwhile, due to the narrow processing window and high strain rate sensitivity of titanium alloys, the fraction of DRX was found to be strongly influenced by the strain rate in some studies [4,7,21]. Therefore, detailed studies of the PDRX behavior of titanium alloys at different strain rates are quite necessary.

In this paper, the effect of the strain rate on the PDRX behavior of TC18 alloy was systematically investigated by thermal simulation compression experiments and the potential mechanism of annealing condition on PDRX was revealed. Furthermore, the kinetic model of recrystallization for TC18 alloy was established based on the DRX and PDRX theory. The changes in grain boundaries, grain size, grain orientation spread (GOS), geometrically necessary dislocations (GNDs) density, etc., during PDRX were characterized and investigated. 

## 2. Experiments Materials and Procedure

### 2.1. Materials

The material studied in this paper is TC18 alloy, the actual chemical compositions (wt.%) are as follows: 5.20Al, 4.92Mo, 4.96V, 1.05Cr, 0.96Fe and balance Ti. The phase transition temperature of the alloy is approximately 875 ± 5 °C. Prior to the isothermal compression, the alloy cut from the forged bars was treated in a solid solution at 900 °C for 30 min to obtain the single β phase microstructure, followed by water quenching. The initial microstructure of the sample is shown by the inverse pole figure (IPF) map in Figure 1a, which consists of equiaxed β grains. It illustrates the distributions of low-angle grain boundaries (LAGBs, 2–15°, white lines) and high-angle grain boundaries (HAGBs, >15°, black lines) in the primitive microstructure. Figure 1b shows the β grain size distribution, and the average initial grain size is about 253 µm. 

### 2.2. Experimental Methods

The cylindrical specimens of 10 × 15 mm used for the compression tests were cut from the solution-treated billet and then mechanically polished. The isothermal compression experiment was performed on a Gleeble 3180 thermos-mechanical simulator. Figure 2 shows the detailed procedure of the hot compression test. Following the heating at 5 °C/s, the specimens were held at the deformation temperature for 5 min to ensure a uniform temperature distribution within the specimen before deformation. The deformation temperature was 900 °C and strain rates ranged from 0.001 to 1 s^−1^. Then, the specimens were compressed to the same deformation degree (50%), followed by annealing for 0–600 s and quenching. To eliminate as much friction as possible generated during compression, the graphite plate was placed between the cylindrical sample and the clamping mold.

### 2.3. Microstructure Characterization

Following the compression test, the samples were cut in half along the compression direction and subjected to microstructure characterization. Considering the non-uniform strain distribution in the deformed specimens, the central region of the specimen with the higher strain was chosen as the observation region. To ensure that the surface quality of the specimens meet the requirements of the EBSD analysis, the cut sections were first pre-polished with silicon carbide sandpaper and then electrolytic polished with the mixed solutions of 300 mL CH_3_OH, 175 mL CH_3_(CH_2_)_3_OH, 25 mL HClO_4_ acid at a voltage of 60 V in a low-temperature environment. EBSD characterizations were carried out on a Helios Nanolab G3 UC scanning electron microscope. The scanned step length is 0.2 µm at 15 kV of accelerating voltage and an accelerating current of 13 nA for EBSD. TSL-OIM software was used to analyze the EBSD data of this work and to perform quantitative statistics. Recrystallized grains were distinguished by the grain orientation spread (GOS) method [20,21,22]. The geometrically necessary dislocation density (GND) was analyzed and quantitatively counted by MTEX-5.8.1 software [23,24].

## 3. Results

### 3.1. Dynamic Recrystallization Behavior during Thermal Deformation

IPF, GOS, and GND maps of the microstructure for TC18 alloy under different strain rates are shown in Figure 3. GOS is calculated by means of the average deviation between the orientation of each point within the grain and the grains’ average orientation. As a result, the recrystallized grains can be distinguished from the deformed matrix according to the GOS index, and the percentage of recrystallization can be determined. Compared to the deformed grains, the recrystallized grains have lower GOS values, which are empirically considered to be <2 [25,26]. At the same time, the distribution of the dislocation density during this process can be visualized by GND diagrams, which are a reliable indication of the level of stored energy. Nye et al. [27] proposed the existence of GND to explain the plastic deformation mode. In combination with the corresponding GOS and GND maps, the specific microstructure evolution under different conditions can be determined. At the low strain rate of 0.001 s^−1^ (Figure 3a–c), larger DRX and irregular elongated grains with abundant sub-grains are identified in the microstructure. Grain boundaries bow out towards areas of high dislocation density and there is a large number of sub-grains within β-deformed grains, which demonstrates that the recovered and recrystallized grains have sufficient time to occur under this condition. With the strain rate increasing to 0.01 s^−1^ (Figure 3d–f), the fraction and size of equiaxial DRX grains are reduced. Meanwhile, the grain boundaries are serrated and DRV is reduced in this condition, characterized by the aggregation of most dislocations at deformed grain boundaries and the reduction of sub-grain boundaries. Small DRX grains occur at some trigonal grain boundaries. Further increasing the strain rate to 0.1 s^−1^ (Figure 3g–i), deformed grains are wider and seriously-elongated, and grain boundaries tend to be flattened. Few small equiaxial grains (showing low GND) locate at the serrated grain boundaries of the uniformly deformed specimen. At high strain rates of 1 s^−1^ (Figure 3j–l), the degree of grain boundary flattening is enhanced and exhibits a relatively high dislocation density. Some of the grain boundaries become serrated with the generation of sub-grains in β grains, and a few tiny equiaxial grains appear at the grain boundaries. The GND is relatively high in the elongated grains but is low in the tiny equiaxial grains (Figure 3l), suggesting the occurrence of DRX consumes the surrounding dislocation, and also indicating a severe degree of plastic deformation or high dislocation density in this region [28]. In addition, DRX grains are generated and grow up in places with large grain boundary undulations and at triple grain junctions, which is also known as discontinuous dynamic recrystallization (DDRX).

Figure 4 quantifies the DRX fraction as well as the percentage of LAGBs at different strain rates. As the strain rate increases from 0.001 to 1 s^−1^, the dynamic recrystallization fraction decreases from 3.7% to 0.6%. Due to the reduction of deformation time, DRX is too late to nucleate and grow. However, LAGBs, as high-energy grain boundaries, to some extent, also reflect the magnitude of deformation energy storage. With the strain rate increasing from 0.001 to 0.1 s^−1^, the percentage of LAGBs exhibits an opposite trend, increasing from 76.7% to 84.3%.

### 3.2. Post-Dynamic Recrystallization Behavior during Post-Deformation Annealing

#### 3.2.1. Effect of the Historical Strain Rate

Figure 5 shows IPF, GOS and GND maps of the microstructure for TC18 alloy after deformation at different strain rates and annealing for 600 s at 900 °C. Following the deformation at low strain rates of 0.001 s^−1^ and annealing (Figure 6a–c), the microstructure is still dominated by elongated β grains, and there are obvious misorientations and LAGBs in the internals of these grains, indicating that static recovery is the main mechanism for this condition. However, there are also a small number of recrystallized grains, approximately 80 µm of grain size at the serrated grain boundaries. Increasing the strain rate to 0.01 s^−1^ (Figure 6d–f), the elongated β grains are reduced and some recrystallized grains are observed to grow abnormally (>200 µm) in the annealing microstructure. A similar phenomenon was found in the study of Fan [29], which referred to the abnormal growth of grains during post-deformation annealing as MDRX, which is based on DRX grains for growth. When the strain rate increases to 0.1 s^−1^ (Figure 6g–i), the fraction of the recrystallization grains is sharply increased. However, elongated deformed grains still exist, and it is obvious that the GND values inside the deformed grains are relatively higher than that in the equiaxial recrystallized grains. However, some thick grains have slightly-elongated morphology and contain a small proportion of sub-grain boundaries formed by entangled dislocations (Figure 6i), demonstrating that SRV plays a key role when extensive SRX proceeds. As for the deformation at 1 s^−1^ (Figure 6j–l), further increases are observed in the fraction of recrystallization, along with an improvement in the homogeneity of the microstructure. Meanwhile, an interesting phenomenon is noted that recrystallized grains are found inside the deformed grains, which may be due to the gradual rotation of deformed sub-crystals during annealing, resulting in the migration of LAGBs to HAGBs.

Figure 6 quantifies the recrystallization fraction and the proportion of LAGBs, the relation of recrystallization fraction and the ratio of LAGBs exhibits similar patterns in other studies [7,11,30]. Annealing for the same time after deformation, as the strain rate increases from 0.001 to 0.01 s^−1^, increases the recrystallization fraction from 10.5% to 20.2%. The recrystallization fraction increases greatly at high strain rates (>0.01 s^−1^), and it increases to 62.2% and 79.7%, respectively, when the strain rate increases to 0.1 and 1 s^−1^. At the same time, the fraction of LAGBs decreases from 85.6% to 60.3%. LAGBs are converted to HAGBs gradually in this process. In addition, comparing the recrystallization fraction at the end of deformation, the recrystallization fraction during annealing increases as does the strain rate.

To further investigate the effect of the strain rate on the PDRX behavior of TC18 alloy, the variation of GND density after deformation and annealing was quantitatively analyzed. According to Nye’s theory, which relates the rotational gradient within the Burgers circuit to the stored GND content, the GND density can be determined by the EBSD data [31,32]. It can be explained as [33]:(1)$$\rho_{GND}=3KAM_{ave}/vb$$
where *b* is the Burgers vector, *v* is the step size taken for EBSD. *KAM_ave_* denotes the mean misorientation in the core region. The GND density of TC18 alloy under different deformation conditions is calculated by the MATLAB program and the corresponding results are displayed in Table 1. The GND density is related to the strain rate during thermal deformation, with the maximum of 2.03 at 0.1 s^−1^, and the minimum of 1.10 at 0.1 s^−1^. This difference comes from the fact that the change in the strain rate affects the onset time of the dynamic recrystallization. The relationship between the GND density and strain rate in the annealing stage exhibits the opposite trend to that in the deformation stage, with the maximum of 1.05 at 0.001 s^−1^, and the minimum of 0.29 at 0.1 s^−1^. Dislocations, as a type of crystal defect, produce lattice disturbances in their vicinity in the form of strain. The increase in lattice strain leads to an increase in energy storage in the deformed material. Therefore, the dislocation density of the deformed matrix affects the driving force of post-recrystallization during subsequent annealing. High dislocation density facilitates recrystallization nucleation and growth, as shown in Figure 5. The consumption of GND density during the annealing process corresponds to the change in recrystallization fraction at different strain rates, so that ΔGND is minimized at 0.001 s^−1^.

#### 3.2.2. Effect of Annealing Time

To better understand the phenomenon of inhomogeneous recrystallization grain size under the deformation with the strain rate of 0.01 s^−1^ and subsequent annealing at 900 °C, microstructure evolution with annealing time was studied, as shown in Figure 7. The microstructure characteristics after deformation have been shown in Figure 3d,e. When the deformed specimen is held at 900 °C for 60 s after deformation, the recrystallization fraction remains at about 3% and the grain size does not change significantly (Figure 7a,b). This indicates that post-dynamic recrystallization has not been initiated at this condition. Extending the annealing time to 300 s, recrystallized grains with different sizes appear around the deformed matrix, and the overall recrystallization fraction increases to 9.8%. This type of larger grain size can be distinguished by OIM analysis software for the PDRX orientation, which after practical analysis can be determined by an orientation difference of 0–0.5°, as shown in Figure 8a,b. The same phenomenon is also seen in the results of other studies, which refer to this recrystallization of larger grains during the annealing process as MDRX, i.e., DRX during deformation grain growth without a gestation period occurs during subsequent annealing and there is a significant gap in the grain size. This recrystallization grain growth mechanism is also referred to as the MDRX mechanism.

## 4. Discussion

### 4.1. Post-Dynamic Recrystallization Mechanism

The microstructure of the metal materials exhibits strong heritability during successive thermomechanical processing. Furthermore, the DDRX grains produced by thermal deformation are usually unstable at high temperatures, leading to the occurrence of MDRX during the subsequent annealing process, as the gray grains in Figure 8b. Ding et al. [20,34] distinguished between the DRX and MDRX grains through misorientation distribution. MDRX regards the previously generated DDRX grains as nuclei, which implies that MDRX no longer requires the nucleation step of conventional static recrystallization and grows rapidly in a short period of time. As a result, the grain size of MDRX grains is relatively larger than that of SRX grains during short-time annealing. For titanium alloys, the fraction of DRX during deformation in the single-phase region generally does not exceed 20%, due to the high stacking fault energy of the β phase. A high dislocation density still exists in the deformed matrix, especially when increasing the strain rate (Figure 3). In the red enlarged area in Figure 8a, it can be observed that G1 contain sub-crystals within them, with a large number of small angular grain boundaries collecting at the sub-crystal boundaries and gradually forming new recrystallized grains, which pointed out that the accumulation of dislocations can act as a driving force for the sub-grain rotation [35]. The polar figure of G1, shown in Figure 8c, clearly illustrates the orientation relationship between them. The crystallographic orientation and color of these sub-grains are highly similar and show a gradual change. The black line L1 illustrates the path where the deformed grains are split into SRX grains by lattice rotation during annealing after isothermal compression, which is a typical CSRX feature. Thus, these microstructural evolutions support the occurrence of CSRX through dislocation transitions, static reversion to SRX, as well as sub-grain transitions merging from LAGBs to HAGBs.

In addition, small SRX grains can be observed to exist at the trigonal mouths of the initial grains, such as G2 in the blue enlarged area in Figure 8a. The HAGB of the curvature at larger deformation is bent toward the adjacent grains by the grain boundary-induced migration mechanism, so that the grain boundaries are bowed out from the low density to the high density and become stable interfaces after reaching a certain size, forming new small SRX grains. This fully exhibits the DSRX characteristics [36]. In addition, the dislocation relationship can further illustrate the DSRX mechanism, as shown in Figure 8d, the dislocation relationship between the grains and G2 in the polar figure proves that the DSRX grains have a random orientation. The red line L2 also proves that the lattice of G2, compared with surrounding grains, has changed completely. In conclusion, it was shown that CSRX and DSRX are the main static recrystallization mechanisms in TC18 titanium alloy, which has also been reported in other titanium alloys [37].

Figure 9 illustrates the schematic diagram of the recrystallization mechanism during deformation and subsequent annealing. On the one hand, dislocations pile up near the grain boundaries, causing large distortion energy at the grain boundaries and promoting recrystallization nucleation during the deformation in the single-phase region. As the grain boundaries migrate, the grain boundaries produce bulges and gradually form DDRX. On the other hand, with the increase of strain, high dislocation density regions are also formed inside the grain. A large number of sub-grains are generated and accompanied by rotation. At the end of deformation and at the beginning of annealing, DDRX grains from the deformation stage absorb the surrounding distortion energy and grow rapidly, forming MDRX. With the extension of annealing time, driven by strain energy and interfacial energy, small DSRX grains are formed, which are distinguished from DDRX. At the same time, sub-grains inside the deformed grains rotate further and LAGBs gradually migrate to the HAGBs, forming CSRX. Finally, with a further extension of time, the dislocations annihilate and the deformed grains all become recrystallized grains under the action of different recrystallization mechanisms.

### 4.2. Recrystallization Kinetic Model

The strain rate affects not only the occurrence of DRX during deformation (Figure 3), but also the driving force of PDRX during annealing (Figure 5). In order to quantitatively describe the effect of the strain rate on recrystallization during successive deformation and annealing, a new recrystallization kinetic model is proposed. Recrystallization fractions of the specimens with different strain rates during the annealing process were supplemented, as shown in Table 2. The total recrystallization fraction is equal to the sum of DRX and PDRX [38];
(2)$$X_{total}=X_{DRX}+X_{PDRX}$$where $X_{DRX}$ is the fraction of dynamic recrystallization at the end of deformation, $X_{PDRX}$ is post-dynamic recrystallization during annealing.

The DRX kinetic model generally established the connection between the DRX fraction and strain by strain–stress curves in most studies. Zhou et al. [39] improved the Avrami equation and coupled strain and strain rate describing the fraction, and the equation is as follows:(3)$$X_{DRX}=1-\exp\left[-\beta_{d}{\left(\frac{\varepsilon -\varepsilon_{c}}{\dot{\varepsilon }}\right)}^{n}\right]$$
where $\dot{\varepsilon }$ is the strain rate; ε is the deformation document; $\varepsilon_{c}$ is the critical strain that can be obtained by the process hardening rate from the stress–strain curve (Figure 10a,b); $\beta_{d}$, *n* are material constants. Taking the natural logarithm for both sides of the equation, *n* can be evaluated from the slope of the linear fit of ln[−ln(1 − X_DRX_)] and ($\frac{\varepsilon -\varepsilon_{c}}{\dot{\varepsilon }}$). Similarly, *β_d_* can be calculated from the intercept in Figure 11a. Meanwhile, the critical strain varies with the strain rate. According to the model of Sellars [40], the critical strain at different strain rates can be expressed by the following equation:(4)$$\varepsilon_{c}=A{\left[\dot{\varepsilon }\exp\left(\frac{Q}{RT}\right)\right]}^{k}$$
where *A* and *k* are material constants; *Q* denotes the deformation energy during the deformation process, which can be determined as 188 kJ/mol in the research of Shi [7]; R is the gas constant (8.31 J/mol/K); T is the deformation temperature (K). To simplify the calculation by going to the natural logarithm on both sides of the formula, *A* and *k* can be obtained according to the linear fit of $ln\varepsilon_{c}$ and $ln\dot{\varepsilon }$ (Figure 11b). The DRX kinetic model is acquired.

During the annealing process, the main recrystallization mechanism is PDRX. As the annealing time extends, the recrystallization fraction is increased by consuming more deformation energy. Therefore, the fraction of PDRX at different strain rates can be calculated by Table 2 with the increase of annealing time, and the PDRX kinetics model can be constructed using the JMAK equation [41,42], as follows:(5)$$X_{PDRX}=1-\exp\left[-\beta_{p}t^{m}\right]$$
where *t* is the annealing time. *m*, $\beta_{p}$ are material-related constants and obtained by calculating the average value under different strain rates in most studies. However, in this work, it was found that the growth rate of the PDRX fraction is closely related to strain rates. The kinetic index *m* and the coefficient $\beta_{p}$ in Equation (5) are strongly related to the strain rate, as shown by the following equation:(6)$$m=A_{1}\ln\left(\dot{\varepsilon }\right)+c_{1}$$
(7)$$\beta_{p}=A_{2}\exp(\frac{-\ln\left(\dot{\varepsilon }\right)}{A_{3}})+c_{2}$$

*A*_1_, *A*_2_, *c*_1_ and *c*_2_ are constant about the material. Therefore, a link between the PDRX fraction and strain rate can be established through *m* and $\beta_{p}$ as a way to explain the PDRX behavior. Next, material constants, such as *A*_1_, *A*_2_, *c*_1_ and *c*_2_ can be obtained by combining the fit to Equations (5)–(7), as shown in Figure 11c,d.

In summary, the parameters solved in the above equation are shown in Table 3, and the recrystallization fraction equation for annealing at different times at different strain rates can be shown by the following equation:(8)$$X_{total}=1-\exp\left[-\beta_{d}{\left(\frac{\varepsilon -A{\left[\dot{\varepsilon }\exp\left(\frac{Q}{RT}\right)\right]}^{k}}{\dot{\varepsilon }}\right)}^{n}\right]+1-\exp\left[-(A_{2}\exp(\frac{-\ln\left(\dot{\varepsilon }\right)}{A_{3}})+c_{2})t^{A_{1}\ln\left(\dot{\varepsilon }\right)+c_{1}}\right]$$

To verify the accuracy of the recrystallization kinetic model, the recrystallization fractions were calculated for different times of annealing at different strain rates, and then the calculated values were compared with the predicted values in Figure 12. Standard statistical parameters, such as the correlation coefficient (*R*) can be expressed as follows:(9)$$R=\frac{\sum_{i=1}^{N}\left(E_{i}-\bar{E}\right)\left(P_{i}-\bar{P}\right)}{\sqrt{\sum_{i=1}^{N}{\left(E_{i}-\bar{E}\right)}^{2}\sum_{i=1}^{N}{\left(P_{i}-\bar{P}\right)}^{2}}}$$
where *E_i_* and *P_i_* are the experimental and predicted recrystallization fraction, respectively. $\bar{E}$ and $\bar{P}$ are the mean values of *E_i_* and *P_i_*. *N* is the total number of data points used in the study. The linear correlation between experimental and predicted curves is defined by *R*, which ranges from −1 to 1. The correlation would be better if the value is closer to 1. Here, R is computed as 0.978, indicating the high prediction accuracy of the developed constitutive model. Therefore, the recrystallization kinetic model is able to predict the recrystallization fraction at different strain rates very well.

## 5. Conclusions

The dependence of DRX and PDRX on the strain rate for TC18 alloy was investigated and the corresponding recrystallization model was also constructed during the post deformation annealing process. The main findings are summarized as follows:(1)During thermal deformation in the single-phase region, DRV is the dominant mechanism. As the strain rate increases from 0.001 s^−1^ to 1 s^−1^, the DRX fraction decreases from 3.7% to 0.6%. The GND density in the deformed matrix is closely related to the strain rate, the higher the strain rate, the higher the GND density;(2)PDRX exhibits a distinct strain rate dependence. The variation of the PDRX proportion at different strain rates is accompanied by the variation of the GND density on the β matrix, indicating that PDRX occurs by absorbing the surrounding deformation storage energy;(3)Deformed and DRX grains undergo different mechanisms during the post-deformation annealing process. MDRX occurs rapidly through the preferential growth of DRX without the process of nucleation. The amount of MDRX is determined by the fraction of DRX during the deformation stage. Subsequently, DSRX by bulging of the grain boundaries and CSRX by sub-grain rotation emerges;(4)Based on the conventional JMAK kinetic equations, a new recrystallization model coupling DRX and PDRX during continuous deformation and annealing is proposed. The correlation coefficient (R) of the model is 0.978. Therefore, the model can accurately describe the correlation between the recrystallization and strain rate for TC18 alloy.

## Figures and Tables

**Figure 1 materials-16-01140-f001:**
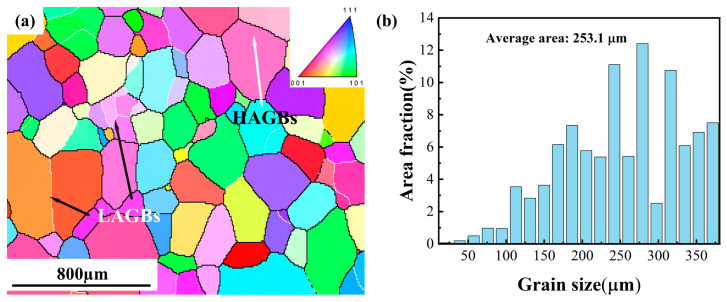
Initial microstructure of the TC18 alloy: (**a**) IPF map; (**b**) grain size distribution.

**Figure 2 materials-16-01140-f002:**
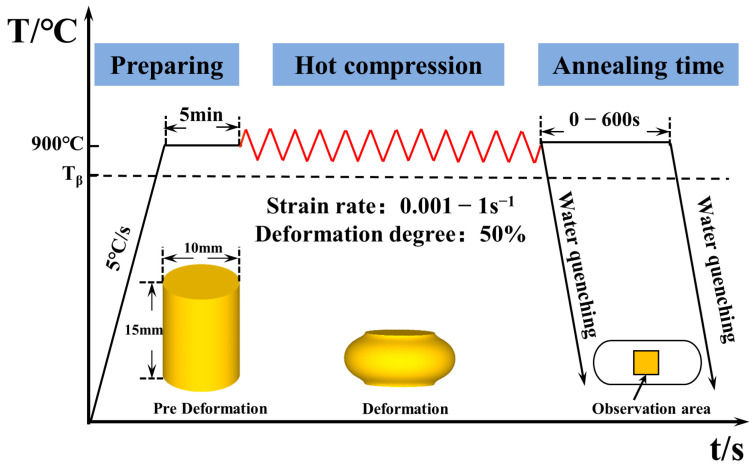
Experimental procedure of the thermal compression experiment.

**Figure 3 materials-16-01140-f003:**
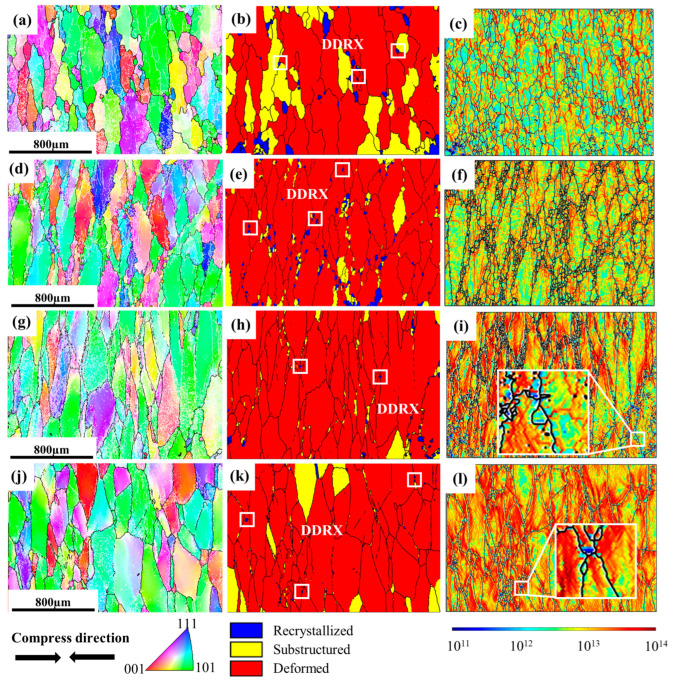
IPF, GOS and GND maps of the specimens deformed at different strain rates: (**a**–**c**) 0.001 s^−1^; (**d**–**f**) 0.01 s^−1^; (**g**–**i**) 0.1 s^−1^; (**j**–**l**) 1 s^−1^.

**Figure 4 materials-16-01140-f004:**
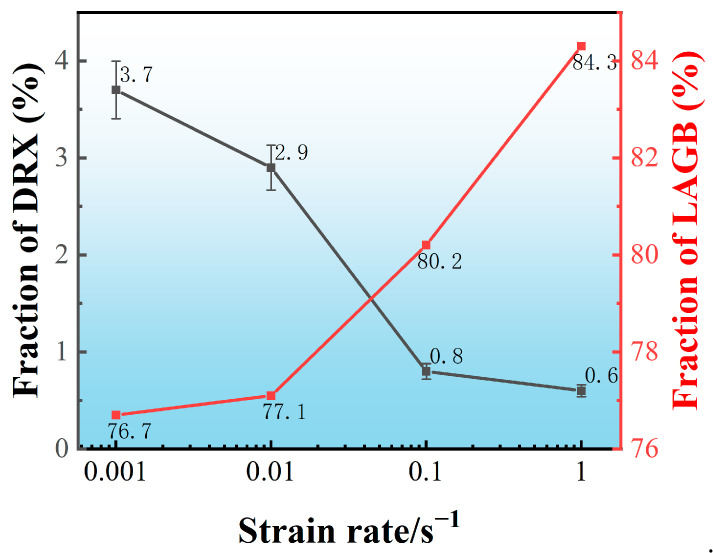
Quantitative analysis of the DRX and LAGB fraction of the specimens deformed at different strain rates.

**Figure 5 materials-16-01140-f005:**
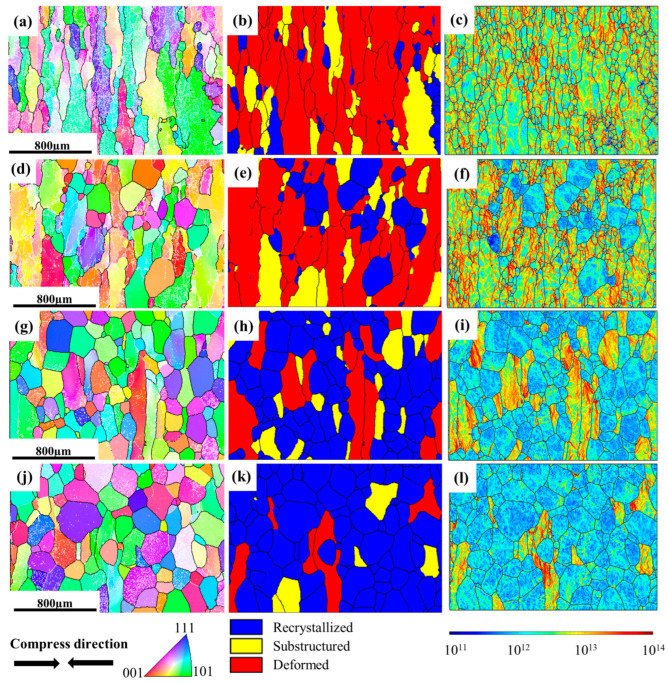
IPF, GOS and GND maps of the specimens deformed at different strain rates and then subjected to annealing for 600 s: (**a**–**c**) 0.001 s^−1^; (**d**–**f**) 0.01 s^−1^; (**g**–**i**) 0.1 s^−1^; (**j**–**l**) 1 s^−1^.

**Figure 6 materials-16-01140-f006:**
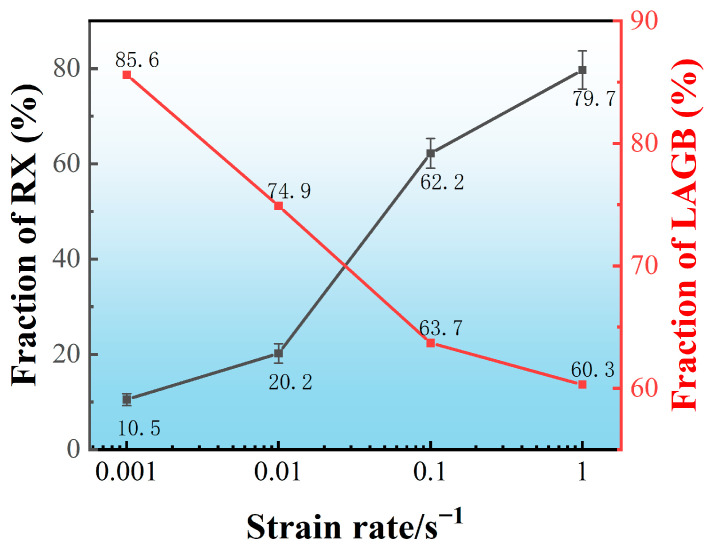
Quantitative analysis of the RX and LAGB fraction of the specimens deformed at different strain rates and then subjected to annealing for 600 s.

**Figure 7 materials-16-01140-f007:**
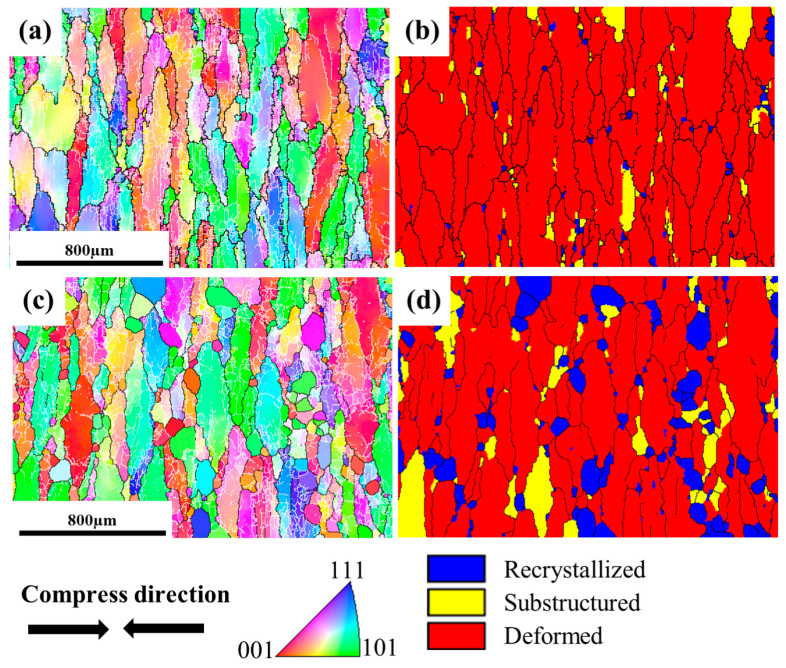
IPF and GOS maps of the specimens subjected to annealing for different times after deformation of 50% with the strain rate of 0.01 s^−1^: (**a**,**b**) 60 s; (**c**,**d**) 300 s.

**Figure 8 materials-16-01140-f008:**
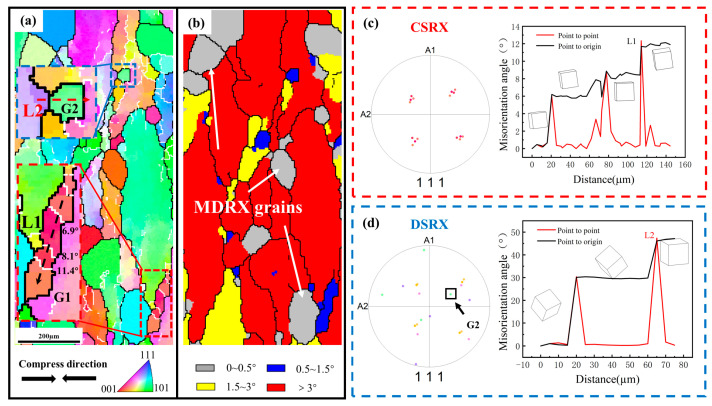
PDRX mechanism of TC18 alloy: (**a**) IPF map; (**b**) GOS map; (**c**) {111} pole figure of G1 and misorientation change along L1; (**d**) {111} pole figure of G2 and misorientation change along L2.

**Figure 9 materials-16-01140-f009:**
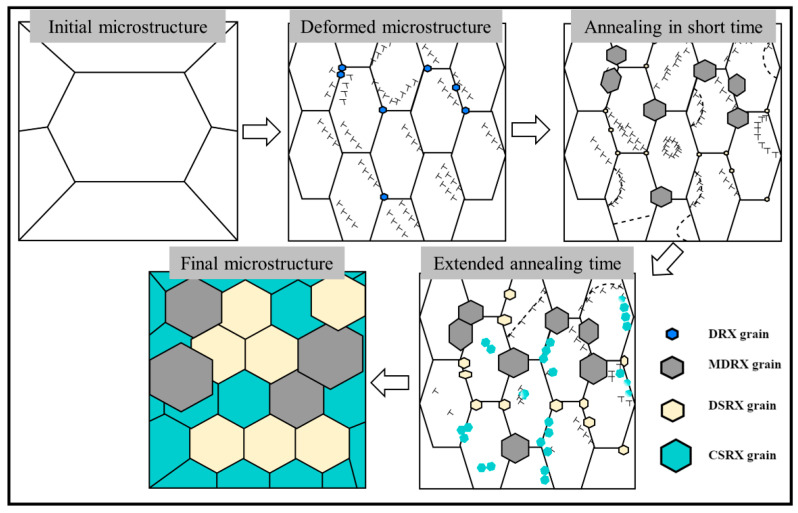
Schematic diagram of the post-dynamic recrystallization of TC18 alloy.

**Figure 10 materials-16-01140-f010:**
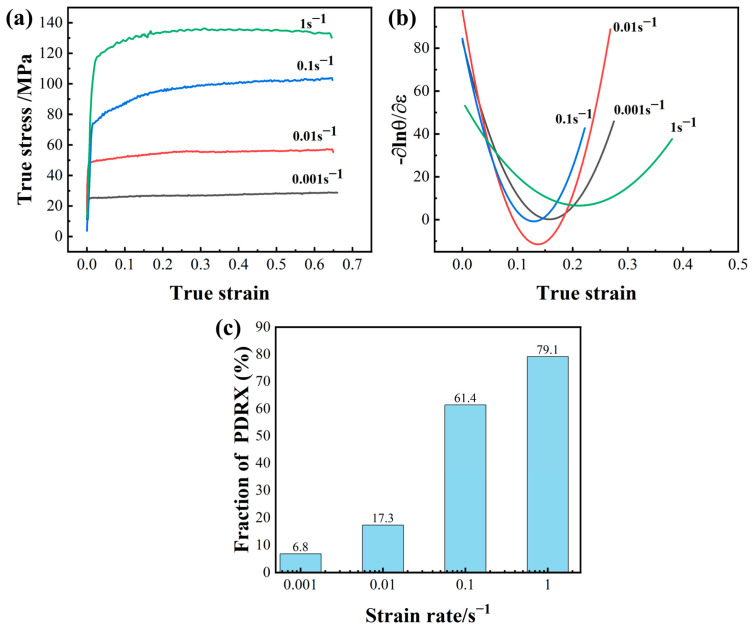
(**a**) Stress–strain curves at different strain rates; (**b**) Critical strain at different strain rates; (**c**) The fraction of PDRX at different strain rates.

**Figure 11 materials-16-01140-f011:**
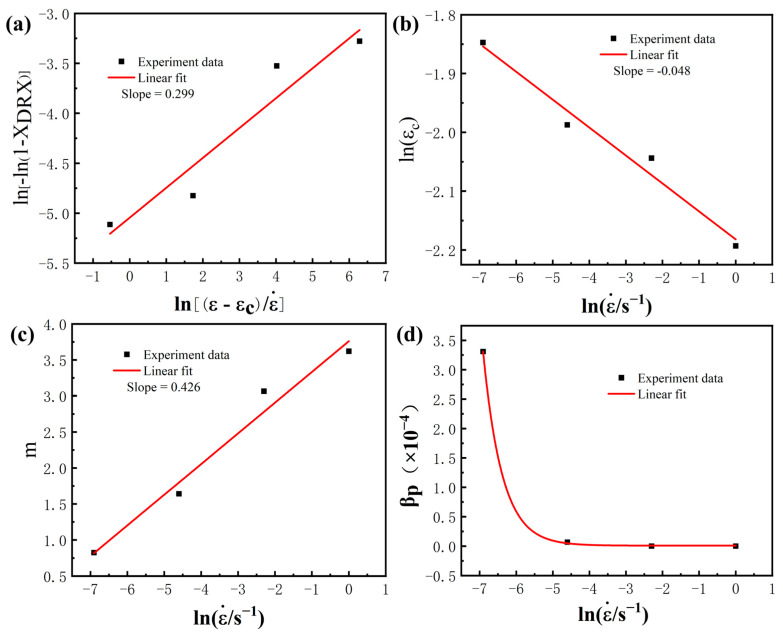
The fitting curves of (**a**) ln[−ln(1−*X_DRX_*)] and ($\frac{\varepsilon -\varepsilon_{c}}{\dot{\varepsilon }}$); (**b**) ln$\varepsilon_{c}$ and ln$\dot{\varepsilon }$; (**c**) n and ln$\dot{\varepsilon }$; (**d**) $\beta_{P}$ and ln$\dot{\varepsilon }$.

**Figure 12 materials-16-01140-f012:**
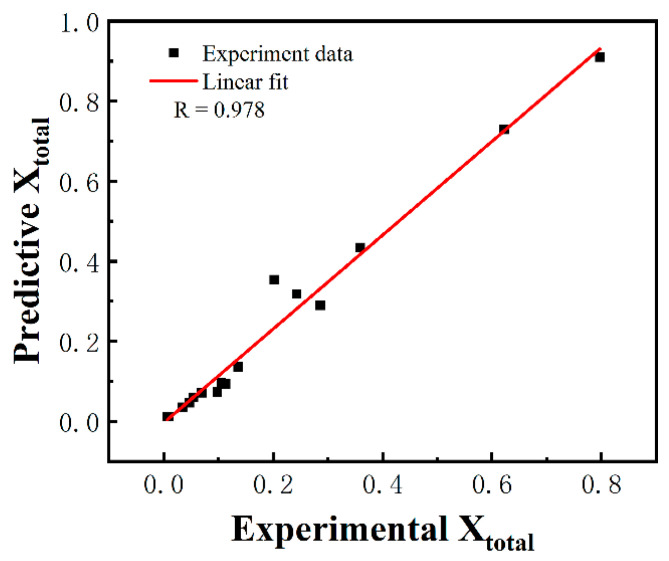
Comparison between the predicted and tested values of the recrystallization fraction.

**Table 1 materials-16-01140-t001:** Average GND density of the deformed titanium alloy under different conditions.

$\dot{\varepsilon }$/s^−1^	Deformation Degree/%	Average GND Density (×10^13^)		
		Immediate Quenching	Annealing for 600 s	ΔGND
0.001	50	1.10	1.05	0.05
0.01	50	1.90	1.04	0.86
0.1	50	2.01	0.49	1.52
1	50	2.03	0.29	1.74

**Table 2 materials-16-01140-t002:** The fraction of DRX and total recrystallization under different strain rates.

$\dot{\varepsilon }$/s^−1^	*X_DRX_*/%	*X_total_* at Different Times/%			
		60 s	180 s	300 s	600 s
0.001	3.7	4.7	5.4	6.9	10.5
0.01	2.9	3.4	9.8	13.6	20.2
0.1	0.8	0.9	11.3	24.3	62.2
1	0.6	0.68	28.6	35.9	79.7

**Table 3 materials-16-01140-t003:** The determined values of the material parameters.

n	$\beta_{d}$	A	k	A_1_	C_1_	A_2_	A_3_	C_2_
0.299	6.45 × 10^−3^	0.283	−0.048	3.76	0.426	5.8 × 10^−10^	0.521	5.28 × 10^−10^

## Data Availability

All raw data supporting the conclusion of this paper are provided by the authors.

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
