# Peer review of "Strain Rate Dependence and Recrystallization Modeling for TC18 Alloy during Post-Deformation Annealing"

_materials, 2023, doi:10.3390/ma16031140_

Round 1
Reviewer 1 Report
I have read the manuscript titled" Strain rate dependence and modeling of recrystallization for TC18 alloy during post-deformation annealing," submitted to Materials Journal for possible publication. I feel the manuscript is well-written and needs only minor revision. I have attached my comments.
I wish authors all success with this paper

Author Response
Thank you for your valuable comments on our article. All your comments are both professional and valuable, which are great significance to improve my scientific research value and the quality of essay writing. The following are my response to your comments one by one. Please see the attachment.

Reviewer 2 Report
1. EBSD is excellent tool for such study. Is it possible to add the estimation of error for determination of different parameters (grain size, fracture of recrystallized grains, dislocation density etc…)?
2. As for the model, presented in the paper it is not clear what is the difference with previous one. The mathematical treatment based on the set of Avrami like equations is quite known and it is actively used [see e.g. A.M. Elwazri, E. Essadiqi, S. Yue. Kinetics of metadynamic recrystallization in microalloyed hypereutectoid steels // ISIJ International. 2004. V. 44, N. 4, P. 744–752.] or in simplified form [Dub, V., Churyumov, A., RodinA., Belikov, S., Barbolin, A. Prediction of grain size evolution for low alloyed steels (2018) Results in Physics, 8, pp. 584-586. DOI: 10.1016/j.rinp.2017.12.028]. What is real motivation to develop a new one and can the results obtained by different models be compared?
3. Presented schematically the stages of new grain growth looks interesting. Is it possible to determine (or estimate) the number of new grains? According the presented data, it must be connected with grain size. It must be key parameter to find the final grain size.
4. The dimension for dislocation density must be indicated in the text, in the pictures (e.g. Fig 3 and 5) and in the tables.
5. Some part of text is difficult to read. E.g. the text in 3.1 describing the Fig.3 is combination of description of picture, comparison with literature and definitions. As a results the long sentences with different comments in or without brackets written. It must be systemized and structured to do it more “readable”.
Author Response

(The authors gave the same response as above.)

Reviewer 3 Report
The paper shows good results to the area of research, specifically Titanium alloys. The kinetic model is well presented and discussed. It is a good scientific paper.
I consider that this study is fundamental to understand the recrystallization mechanisms to titanium alloys under different strain values, so this is an original investigation in this field.
As it provides specific values about the relationship strain rate-recrystallization extent for the TC18 titanium alloys not available in the scientific literature.
Specific values about the relationship strain rate-recrystallization extent for the TC18 titanium alloys.
The methodology performed was sufficient to find the relationship strain rate-recrystallization extent for the TC18 alloy at rather low strain rates. But of course this study could be extended to explore the effect of higher values of strain rate. However, this implies a hugh investigation that probably lay outside the scope of the paper itself.
It is enough to the data considered to build the paper.
I find suitable the proposed conclusions, as the values of some important parameters such as DRX fraction, GND density, are provided. Also, they addressed a recrystallization model coupling DRX and PDRX during continuous deformation and annealing of the TC18 alloy.
They used important references to make a well discussion of their main findings and measurements.
Author Response

(The authors gave the same response as above.)
